# Lipidomic Profile of Human Sperm Membrane Identifies a Clustering of Lipids Associated with Semen Quality and Function

**DOI:** 10.3390/ijms25010297

**Published:** 2023-12-25

**Authors:** Andrea Di Nisio, Luca De Toni, Iva Sabovic, Alessia Vignoli, Leonardo Tenori, Stefano Dall’Acqua, Stefania Sut, Sandro La Vignera, Rosita Angela Condorelli, Filippo Giacone, Alberto Ferlin, Carlo Foresta, Andrea Garolla

**Affiliations:** 1Department of Medicine, University of Padova, 35128 Padova, Italy; andrea.dinisio@unipd.it (A.D.N.); luca.detoni@unipd.it (L.D.T.); iva.sabovic@gmail.com (I.S.); alberto.ferlin@unipd.it (A.F.); andrea.garolla@unipd.it (A.G.); 2Magnetic Resonance Center (CERM) at the Department of Chemistry “Ugo Schiff”, University of Florence, 50019 Sesto Fiorentino, Italy; vignoli@cerm.unifi.it (A.V.); tenori@cerm.unifi.it (L.T.); 3Consorzio Interuniversitario Risonanze Magnetiche MetalloProteine (CIRMMP), 50019 Sesto Fiorentino, Italy; 4Department of Pharmaceutical and Pharmacological Sciences, University of Padova, 35129 Padova, Italy; stefano.dallacqua@unipd.it (S.D.); stefania.sut@unipd.it (S.S.); 5Department of Clinical and Experimental Medicine, University of Catania, 95125 Catania, Italy; sandrolavignera@unict.it (S.L.V.); rosita.condorelli@unict.it (R.A.C.); 6Centro HERA-Unità di Medicina della Riproduzione, Via Barriera del Bosco, 51/53, Sant’Agata li Battiati, 95030 Catania, Italy; filippogiacone@yahoo.it; 7Department of Medicine, Unit of Andrology and Reproductive Medicine, University of Padova, Via Giustiniani, 2, 35128 Padova, Italy

**Keywords:** membranes, fertility, metabolomics, seminolipid, PUFA, cholesterol, reproduction

## Abstract

Reduced sperm motility and/or count are among the major causes of reduced fertility in men, and sperm membranes play an important role in the spermatogenesis and fertilization processes. However, the impact of sperm lipid composition on male fertility remains under-investigated. The aim of the present study was to perform a lipidomic analysis of human sperm membranes: we performed an untargeted analysis of membrane lipid composition in fertile (N = 33) and infertile subjects (N = 29). In parallel, we evaluated their serum lipid levels. Twenty-one lipids were identified by their mass/charge ratio and post-source decay spectra. Sulfogalactosylglycerolipid (SGG, seminolipid) was the most abundant lipid component in the membranes. In addition, we observed a significant proportion of PUFAs. Important differences have emerged between the fertile and infertile groups, leading to the identification of a lipid cluster that was associated with semen parameters. Among these, cholesterol sulfate, SGG, and PUFAs represented the most important predictors of semen quality. No association was found between the serum and sperm lipids. Dietary PUFAs and SGG have acknowledged antioxidant functions and could, therefore, represent sensitive markers of sperm quality and testicular function. Altogether, these results underline the important role of sperm membrane lipids, which act independently of serum lipids levels and may rather represent an independent marker of reproductive function.

## 1. Introduction

Reduced sperm motility and/or count are among the major causes of reduced fertility in men. Regarding somatic cells, sperm cells also present a cell membrane with a phospholipid bilayer, whose composition strongly varies during sperm maturation [1]. In this regard, the composition and packaging of the phospholipid bilayer is recognized to have a strong influence on its biophysical properties and, as a consequence, on cell function [2]. Despite some variations among different mammalian species, mature sperm plasma membranes are characterized by a peculiar composition, with a low content of cholesterol and a high proportion of lipids containing a fatty acid chain [3,4]. In particular, cholesterol, due to its condensed polycyclic structure, is known to reduce the conformational freedom of the phospholipid bilayer, with important consequences on the membrane fluidity, ion channels’ function, and activation of the signaling pathways [5]. Given its importance in regulating sperm function, membrane cholesterol undergoes important changes from early spermatogenesis, when de novo synthesis is massively performed due to enlargement of the male germ cell size and surface remodeling [6], until the late steps of fertilization. In this particular framework, the local high bicarbonate and albumin concentrations of the female tract are invariably associated with the lowering of the membrane cholesterol content and with further sperm membrane reorganization [7]. These changes have been demonstrated as key steps conferring forward motility to sperm cells and preparing membranes for capacitation, the acrosome reaction, and fusion with the oocyte [8,9]. Thus, if the progressive reduction of cholesterol sustains the fertility potential of mature spermatozoa, the levels of membrane cholesterol are expected to lead to the opposite effect [10,11], leading to male infertility [12].

In addition to cholesterol, a major component of sperm membranes is represented by glycolipids [13]. Glycolipids are synthesized through the addition of a glycosidic head group to ceramide [14]. The only exception to this general rule is represented by the seminolipid, or sulfogalactosylglycerolipid (SGG), a peculiar glycolipid characterized by a sulfo-galactosyl glycerol and found exclusively in the membranes of sperm cells, where it plays a well-recognized role in different stages of sperm life, from spermatogenesis to the acrosome reaction and sperm–oocyte fusion [15,16].

In addition to the aforementioned roles of SGG and cholesterol, not only in spermatogenesis but also in membrane stability, it has been observed that a higher degree of phospholipid saturation is associated with reduced semen quality, which is conversely increased by the presence of major poly-unsaturated fatty acids (PUFAs) [17]. Phospholipids are crucial for the optimal function of sperm membranes, and lipids have both a structural and functional role in cells; therefore, their relative abundance and lipid class components play an important role in defining the structure of biological membranes. There is evidence of an exchange of phospholipids between sperm and the surrounding fluid during epididymal transit [18] and changes in the lateral inhomogeneity in the membrane during capacitation [19]. This suggests that the dynamic changes in sperm membranes may play an important role in sperm motility and function [20]. Indeed, reduced sperm motility has been associated with altered serum lipid composition [21], suggesting that blood lipid composition may play a role in optimal sperm function, which is consistent with the negative impact of obesity and dysmetabolism on male fertility [22]. However, the impact of serum lipids on male fertility still remains controversial. The aim of the present study was, therefore, to perform a wide untargeted lipidomic and metabolomic analysis of human sperm membranes in fertile normozoospermic men and infertile patients, together with their serum lipid profiles, in order to investigate the association between sperm and serum lipid composition and their impact on male reproductive potential.

## 2. Results

The clinical and semen characteristics of the fertile subjects and infertile patients are reported in Table 1. Consistent with the homogeneous sampling of the subjects, no significant differences in terms of age or BMI were observed between the two groups. However, the infertile patients showed significantly lower sperm concentrations, viability, and progressive sperm motility than the fertile subjects, but no difference in terms of sperm morphology. From the analysis of the serum hormones, no significant differences were observed in the T, LH, FSH, or estradiol levels. Also, no significant differences were found in the total serum cholesterol, LDL, HDL, or triglycerides (Table 1).

### 2.1. Lipidomic Analysis

After membrane lipid extraction and lipidomic untargeted analysis by LC-MS, we were able to identify twenty-one lipids from the sperm membranes of all subjects (Table 2).

The quantitative analysis of the respective lipid AUC, standardized for the number of cells, allowed for the comparative evaluation of the relative lipid composition of the sperm membranes between the two groups (Figure 1). Sulfogalactosylglycerolipid (SGG, seminolipid) was the most abundant component in the membranes (72%) and was significantly reduced in the infertile subjects (Figure 2; *p* = 0.004). Cholesterol sulfate represented the second most abundant lipid in the sperm membranes (6%) and was more abundant in the sperm membranes of the infertile group (*p* = 0.04). In addition, we observed a significant proportion of PUFAs (63% of the remaining lipids in the membranes), such as C20:4 (arachidonic acid) and phosphatidylcholine (C22:6 16:1), representing 12% and 25% of the total PUFAs in the sperm membranes, respectively. Phosphatidic acid, phosphatidylcholine, and lysophosphatidylethanolamine were all significantly reduced in the infertile group (all *p* < 0.05).

In order to rule out any major effect of seminal plasma debris (immature germ cells, leukocytes), in a subsample of patients (N = 5 from the infertile group and N = 5 from the fertile group), we compared the membrane lipid composition between the Percoll-isolated sperm cells and the pelleted semen samples. No difference was observed between the two cell populations (Appendix A, all *p* > 0.05). Therefore, for further analyses, we considered the data obtained from pelleted semen samples, as reported in Section 4.

Principal component analysis (PCA) was carried out for subsequent statistical analysis to determine the variation in lipids and the possible relationships between their levels and other semen or serum parameters. The PCA results are summarized in Table 3.

The twenty-one lipids were grouped into five PCs, accounting for 74% of the total variance. We then included these clusters as independent variables in a multilinear regression analysis with the following parameters: serum total cholesterol, HDL, LDL, triglycerides, sperm progressive motility, normal morphology, sperm viability, and sperm concentration. Interestingly, none of the sperm lipid clusters was associated with serum lipid markers. Conversely, one of the five clusters (including SGG, phosphatidic acid, and phosphatidylcholine C18:1 16:0) was significantly associated with increased sperm motility, viability, and concentration. A second cluster (including cholesterol sulfate, phosphatydilcholine C14:0 18:1 and C22:6 16:1) was associated with sperm progressive motility. To further characterize these associations, we performed Pearson’s correlation analyses between the components of the two aforementioned clusters and the semen or serum parameters (Figure 2). We found that SGG was positively associated with LH (r = 0.657, *p* < 0.001), FSH (r = 0.469, *p* = 0.001), the sperm concentration (r = 0.326, *p* = 0.010), the total sperm count (r = 0.392, *p* = 0.002), and motility (r = 0.495, *p* = 0.001), and phosphatidylcholine C22:6 16:1 was positively associated with the sperm concentration (r = 0.608, *p* < 0.001), sperm count (r = 0.564, *p* = 0.001), motility (r = 0.610, *p* < 0.001), and viability (r = 0.535, *p* = 0.011). Conversely, cholesterol sulfate was negatively associated with sperm motility (r = −0.282, *p* = 0.04) and sperm morphology (r = −0.317; *p* = 0.025). None of the sperm lipids were associated with the serum lipids (all *p* > 0.05).

To better characterize the association of the membrane lipids with sperm motility, we performed lipid evaluation by LC-MS in a subset of semen samples from the fertile group (N = 10) after performing the swim-up technique to obtain two distinct populations of motile and immotile sperms. Using repeated-measures ANOVA, we identified significant differences between the two fractions of the following lipids: cholesterol sulfate (*p* < 0.001), SGG (*p* = 0.03), phosphatidic acid (C18:0 20:4) (*p* < 0.001), phosphatidylcholine (C18:1 16:0) (*p* = 0.002), phosphatidylcholine (C14:0 18:1) (*p* = 0.046), and phosphatidylcholine (C22:6 16:1) (*p* = 0.009). In particular, the motile fraction of sperm was characterized by an increase in all lipids except cholesterol, which was reduced (Figure 3).

### 2.2. Metabolomic Analysis

To characterize the role of the sperm membrane on sperm motility and sperm count, random forest (RF) models were built using the bucket matrix of CPMG spectra by further comparing the relative importance of metabolites between the infertile patients and the fertile controls. The RF model discriminating the oligozoospermic infertile subjects and the controls showed a significant differential clustering: 74% accuracy, 64% sensitivity, 80% specificity, and 0.71 AUROC (*p*-value = 0.01) (Figure 4A). In contrast, the RF model discriminating controls and the asthenozoospermic patients did not provide significant results: 62% accuracy, 44% sensitivity, 68% specificity, and 0.67 AUROC (*p*-value > 0.05) (Figure 4B). The quantified metabolites were analyzed via univariate statistics to characterize which contributed the most and the most significantly to the differences among the groups. The levels of acetate, glycine, and sn-glycero-phosphocholine were found to be significantly higher in the seminal plasma of the oligozoospermic patients (Figure 4C). Instead, no metabolites emerged as significant in the comparison between the controls and the asthenozoospermic patients, although sn-glycero-phosphocholine appeared to be 1.4-fold higher in the latter (Figure 4D).

Acetate, glycine, sn-glycero-phosphocholine, lysine, aspartate, and glutamine were significantly correlated with the sperm concentration in the whole group (Figure 4E). Moreover, sn-glycero-phosphocholine also showed a statistically significant correlation with sperm motility. Several other correlations emerged from the analysis, but none of them reached statistical significance after FDR correction.

## 3. Discussion

In this study, we performed an untargeted analysis of the lipid composition of human sperm membranes in fertile and infertile subjects. This analysis allowed for the quantification of 21 lipids in 62 semen samples, the levels of which were independent of the serum lipid levels. Important differences emerged between the fertile and infertile subjects, leading to the identification of specific clusters of sperm membrane lipids that were the most significantly associated with the semen parameters. Among these, in particular, cholesterol sulfate, SGG, and PUFAs represented the most important predictors of semen quality. In regards to the lipid profile of the infertile patients, we observed an increase in the contents of saturated and monounsaturated fatty acids together with a decrease in the species containing PUFAs compared to the fertile controls. As a high degree of fatty acid unsaturation is a physiological characteristic of sperm membrane lipids and crucial for the fluidity of the membrane, these results could be explained through the correlation with sperm motility, highlighting the role of oxidative stress in its mechanisms [4]. In parallel, we observed an important increase in cholesterol sulfate in the infertile patients. Cholesterol sulfate represents about 6% of the total cholesterol both in spermatozoa and seminal plasma [23], which is in agreement with our data. It is known that sperm cells undergo an important rearrangement of membranes during epididymal transit and the capacitation process, especially by cholesterol removal, which leads to a more fluid sperm membrane, which is necessary for the acrosome reaction and penetration of the egg cell [24,25]. The progressive reduction of cholesterol in the sperm membrane is a requisite for the acquisition of the fertilization capacity of mature spermatozoa; on the other hand, high levels of membrane cholesterol are expected to have the opposite effect [12]. The regulation of membrane cholesterol during sperm maturation and fertilization is very complex and regulated by many factors [26], but Sertoli cells are recognized as the first step in cholesterol acquisition by germ cells. In vitro, it has been demonstrated that Sertoli cells are able to synthesize cholesterol from acetate [27]. However, this de novo synthesis is not a sufficient source of cholesterol to ensure spermatogenesis in vivo. Therefore, it is necessary to import cholesterol from the blood circulation [26]. Evidence from human studies indicates that male obesity and diet composition could play an important role in the deregulation of spermatogenesis, sperm maturation, or fertilizing ability, but the majority of studies have not considered the plasma lipid profile. The impact of serum lipids on male fertility indeed remains controversial; even if cholesterol is essential for male fertility [28], a recent study reported that hypercholesterolemia is not associated with sperm concentration or motility in men [29]. It has also been shown that there is no correlation between the sperm concentration and the serum total cholesterol or triglycerides in humans [21]. Our data are in agreement with these studies, confirming that the association of sperm cholesterol with spermatogenesis and sperm motility is independent of serum lipids.

Given the elevated number of lipids identified by our untargeted analysis, we performed PCA to identify the lipid clusters mostly associated with semen quality. This analysis has confirmed the important role of cholesterol and PUFAs in sperm motility and fertility potential. It is well known that the fluidity of the sperm membrane is very high due to the unusually high proportion of long-chain PUFAs and that the fluidity and flexibility of the membrane are very important for the acquisition of forward sperm motility [12,30]. Fatty acids are extensively involved in sperm development, maturation, and fertilization events [31]. Many studies have examined the relationship between dietary fatty acids and fertility [32]. In immature germ cells, the percentage of saturated and essential fatty acids is higher, whereas long-chain PUFAs are significantly lower when compared to mature spermatozoa [31]. Interestingly, spermatozoa possess a higher proportion of the most representative PUFA (C22:6 n-3) compared to red blood cells [33]. This suggests that sperm cells have active fatty acid metabolism and are desaturated during spermatogenesis or maturation [34].

Safarinejad et al. [35] compared the composition of sperm fatty acids in fertile and infertile men and found higher omega-3 fatty acids in the fertile group but higher levels and percentages of arachidonic acid in the infertile group. In our study, SGG was the most abundant lipid in sperm membranes and significantly reduced in the infertile patients. SGG is the major anionic glycolipid present only in the sperm plasma membrane of mammals. The importance of SGG for spermatogenesis, as well as for the acrosome reaction and fertilization fusion, is well known [16]. The reduction of SGG in infertile patients, after standardization for the number of cells extracted for LC/MS analysis, confirms the role of SGG as a possible marker of spermatogenesis and fertilization. This hypothesis is confirmed by its positive association with FSH and LH, and its negative correlation with sperm concentration and motility. However, even if the role of SGG in spermatogenesis is well recognized, the molecular mechanisms underlying the action of SGG in this process are still unclear. Moreover, given its role in membrane stability, the positive association with sperm progressive motility is not surprising, as confirmed by a recent study showing a reduction of SGG in asthenozoospermic men [4]. The functional role of SGG in membranes is confirmed by its presence in membrane microdomains in sperm cells. As an ordered lipid, SGG is an integral component of lipid rafts, membrane domains that are platforms of cell interaction and signaling, and like other glycolipids, its involvement in cell adhesion is through its direct affinity for a number of cell surface proteins, extracellular proteins, and extracellular matrix proteins [16]. These properties are fundamental to spermatogenesis, which is greatly dependent on the interaction between developing TGCs themselves as well as between TGCs and Sertoli cells. SGG could therefore contribute to the organization and/or stability of these functional microdomains and, therefore, modulate germ cell function during spermatogenesis or membrane stability and function in mature sperms.

Significant changes in lipid metabolism, choline metabolism, and cholesterol metabolism have been reported in the seminal plasma of infertile patients with asthenozoospermia [36]. Another study performed a metabolomic analysis of fatty acids in the seminal plasma of healthy and asthenozoospermic subjects and identified high levels of oleic and palmitic acid in the samples from infertile men [37]. To this end, we performed NMR analysis on the seminal plasma of the same subjects to characterize the metabolic differences between the infertile patients and the fertile controls.

This is a preliminary study on a limited number of subjects, and further studies on a larger sample number are needed to better define the relationship between lipid composition and sperm function. Moreover, the retention times of the identified lipids should be verified by additional analytic techniques, such as magnetic resonance methods, as the possibility of in-source decay of more complex lipids cannot be ruled out. However, to our knowledge, this is the first report about the determination of untargeted lipidomic and metabolomic analysis in the sperm of infertile and fertile subjects. Importantly, our results are representative of sperm cell populations since no major difference in lipid composition has emerged from the lipidomic comparison of selected sperm cells after isolation by density gradient. Altogether, these results underline the important role of seminal lipids, which act independently of serum lipid levels and could rather represent an independent marker of spermatogenetic and reproductive function. Dietary PUFAs and SGG, in particular, have acknowledged antioxidant functions and could, therefore, represent the most sensitive markers of sperm quality and testicular function. However, to unambiguously associate SGG levels to male infertility, studies on larger populations are needed to provide a threshold level of sperm SGG that segregates between fertile and infertile men. The multi-factorial role of SGG in male reproduction has already been proven in animals and in vitro studies on human sperm, confirming the validity of SGG levels as a marker of male fertility. Such a reliable parameter is much needed at the present time, since the current standard semen parameters have low prediction accuracy for sperm fertilizing ability and male fertility.

## 4. Materials and Methods

### 4.1. Subjects and Semen Analysis

Twenty-nine infertile patients with altered semen parameters were included in the study. The inclusion criterion was an age between 20 and 40 years. The exclusion criteria were varicocele, metabolic syndrome, malignancies, history of cryptorchidism, a history of testicular cancer, and medical treatments or dietary supplements in the 3 months preceding the study. In infertile patients, the female factor was excluded as reported previously [12]. Given the aim of the study, to achieve a representative analysis of the whole sperm population in semen and to avoid possible bias from other cell types, we selected a priori only subjects with non-sperm cells (spermatids, round cells, leukocytes) <1 million/mL. Thirty-three age-matched normozoospermic fertile subjects, who had children in the previous two years, served as controls. All patients underwent physical examination, including anthropometric measurements, such as body weight, height, waist, and hip circumference. Body mass index (BMI) was calculated as the ratio between body weight (in kg) and the square of the height (in meters). All subjects underwent semen donation by masturbation into sterile containers after 2–5 days of sexual abstinence. The samples were allowed to liquefy for 30 min and examined for sperm count, viability, motility, and morphology according to the WHO criteria (WHO, 2021). The semen parameter thresholds, which allowed for further characterization of oligozoospermia and asthenozoospermia, were based on the 5th percentiles from the WHO 6th Edition (2021). After semen evaluation, the samples were washed twice in phosphate-buffered saline (PBS). The cell number of each sperm sample was then counted with a Makler counting chamber, pelleted, and stored at −80 °C until use.

### 4.2. Swim-Up

The isolation of sperm cells with high progressive motility was performed by swim-up procedure, as previously described, with slight modifications [38]. As a brief description, the spermatozoa were washed twice in SWM and pelleted at 700g for 7 min. After the second centrifugation, the sperm cell pellet was covered with 0.5 mL of fresh SWM and incubated at 37 °C for another 45 min. Supernatants containing the sperm fraction with a high percentage of progressive cell motility were then carefully separated from the underlying cell suspension, and both fractions were washed and frozen and then stored at −80 °C until use.

### 4.3. Sperm Cell Isolation by Density Gradient

In order to isolate the sperm populations on the basis of their morphology and motility as well as their separation from leukocytes and immature germ cells, an isotonic density gradient was applied as described elsewhere [39]. To describe it briefly, isotonic 100% Percoll (Sigma Chemical Co., St Louis, MO, USA) was obtained by adding nine parts of Percoll to one part of Earle’s salt solution 10× (Imperial, UK). The 100% Percoll was diluted with Earle’s salt solution to obtain the following dilutions: 30%, 35%, 40%, 45%, 50%, 55%, 60%, 70%, 80%, and 100%. The gradient column was prepared in a 15 mL Falcon tube by gently layering 1 mL of each of the above-mentioned solutions, starting from the 100% fraction at the bottom (0.5 mL for each dilution was used when the semen volume available was 0.5 mL). One milliliter of the semen (or the whole semen in cases with less than 1 mL of ejaculate) was diluted with Earle’s solution (1:2) and centrifuged at 400 *g* for 15 min at 18 °C. The semen cell pellet was resuspended in 0.5 mL of Earle’s solution. The semen cell suspension was gently stratified on top of the discontinuous Percoll gradient and centrifuged for 25 min at 800 *g* at 18 °C. The single Percoll fractions were separated, and each was put into a single test tube. The single fractions were analyzed in order to select the ones with the greatest concentration of immature germ cells. The fractions that contained the majority of immature germ cells (30%, 35%, 40%, and 45%) were mixed with Earle’s solution (1:2) and centrifuged at 150 *g* for 10 min at 18 °C. The pellet was resuspended in 1 mL of Earle’s solution, and the cell concentration was evaluated.

### 4.4. Extraction and Quantitative Determination of Lipids by LC-MS

Analysis of the phospholipids was performed by LC-MS using the Agilent 1260 chromatograph–mass spectrometer, ion trap MS-500 (Varian, Palo Alto, CA, USA). Column Tosohas TSK gel amide 3.0 × 150 was used, and mobile phases were water 0.1% formic acid (A) and acetonitrile (B). The gradient was: min 0, 10:90 (A:B); min 20, 20:80 (A:B); min 25, 20:80 (A:B); min 45, 65:35 (A:B); min 51, 65:35 (A:B); min 52, 10:90 (A:B), and re-equilibration for 10 min. The spectra were acquired in the negative ion mode in the range of *m*/*z* 50–1000. Fragmentation of the main ionic species was obtained using the turbo data-dependent scan (tdds) function of the instrument. Tentative identification of the phospholipids was performed by comparison with the literature. For quantification of the phospholipids, phosphatidyl serine was used as a reference compound, with phosphatidyl inositol and phosphatidyl choline. The samples were prepared by extracting 1 mL of semen in dichloromethane (0.5 mL) in an Eppendorf tube and separating the organic layer. Extraction was repeated three times. The organic layers were evaporated in a vacuum in a round-bottom flask. Finally, the residue was dissolved in 200 microliters of acetonitrile and used for the analysis.

### 4.5. NMR Analysis

The seminal plasma samples were thawed at room temperature and shaken before use. A total of 350 µL of sodium phosphate buffer (75 mM Na_2_HPO_4_x7H_2_O; 20% (*v*/*v*) ^2^H_2_O, 6.1 mM NaN_3_; 4.6 mM 3-(Trimethylsilyl)propionate-2,2,3,3-d4; pH 7.4) was added to 350 µL of each sample. Each mixture was homogenized by vortexing for 30″, and then 600 µL was transferred into a 5 mm NMR tube. The NMR spectra were acquired at a Bruker 600 MHz spectrometer (Bruker BioSpin, Billerica, MA, USA) operating at 600.13 MHz proton Larmor frequency equipped with an automatic and refrigerated (6 °C) sample changer (SampleJet, Bruker BioSpin, Billerica, MA, USA). To ensure high spectral quality and reproducibility, the spectrometer was calibrated daily following the strict standard operating procedures. The detailed description of instrument configuration can be retrieved from our previous publication [40]. All samples were acquired at 310 K.

Seminal plasma consists of various biochemical components, such as cholesterol, lipids, proteins, and metabolites [41]. Thus, to allow for the selective detection of the signals of low-molecular-weight metabolites, a water-suppression 1D 1H CPMG experiment (cpmgpr1d; Bruker BioSpin, Billerica, MA, USA) was performed. The acquisition was performed using 32 scans, 4 dummy scans, 73,728 datapoints, a spectral window of 12,019 Hz, an acquisition time of 3.067 s, a relaxation delay of 4 s, and a total spin-echo delay of 80 ms. A line-broadening factor of 0.3 Hz was applied to each free-induction decay before the application of the Fourier transform. The transformed spectra were corrected for phase and baseline distortions with automatic routines and calibrated at the glucose-6-phosphate signal at δ 5.22 ppm using TopSpin 3.6 (Bruker BioSpin, Billerica, MA, USA).

### 4.6. Sex Hormone Quantification

Blood was collected in the fasting state between 08:00 and 10:00 AM. The serum total testosterone (T), follicle-stimulating hormone (FSH), estradiol (E), and luteinizing hormone (LH) were evaluated by commercial electrochemiluminescence immunoassay methods (Elecsys 2010; Roche Diagnostics, Mannheim, Germany). For all parameters, the intra- and interassay coefficients of variation were <8% and <10%, respectively. All determinations were performed in duplicate.

### 4.7. Biochemical Serum Markers Evaluation

The total blood cholesterol, LDL-cholesterol, HDL-cholesterol, and triglycerides were measured through standard biochemical methods by commercial electrochemiluminescence immunoassay methods (Elecsys 2010; Roche Diagnostics, Basilea, Swiss) at the Central Laboratory of University Hospital, Padova.

### 4.8. Statistical Analysis

The results were expressed as the means ± SDs. The Shapiro–Wilk W test for normality was used to check the distributions of the variables. The Mann–Whitney test was used to assess differences among the groups in the anthropometric, seminal, and hormonal parameters and in the concentrations of the serum and seminal lipids. Adjustment for multiple comparisons was calculated with the Bonferroni–Holm method. The Spearman rank correlation coefficients were calculated to evaluate the correlations between the concentrations of sperm lipids and the variables of interest. Given the elevated number of sperm lipids and their elevated inter-correlations, in order to identify a reduced cluster of lipids associated with semen parameters, we performed principal component analysis (PCA) to reduce the data variability. This is a data-reduction technique aimed at explaining most of the variance in the data by transforming a set of correlated measured variables into a new set of uncorrelated principal components (PCs), which preserve the relationships present in the original data [42]. PCA can be easily extended to the simultaneous analysis of multiple correlated data sets. In the present study, PCA was conducted to assess the possible distribution of the different sperm membrane lipids, as well as to assess any possible correlation with seminal parameters. Correlation coefficients < 0.4 were discarded. Finally, a hierarchical cluster analysis was conducted to confirm some of the conclusions obtained by the PCA. We considered Eigenvalues > 1 as a threshold to select the final clusters, obtaining a final set of 5 clusters that explained 74% of the total variance, with a Kaiser–Meyer–Olkin value = 0.626, *p* < 0.001. The Varimax method with Kaiser normalization was used as a rotation method for the component matrix. For the comparison of immotile and motile sperm fractions within subjects, we performed a repeated-measures ANOVA. For metabolomic analysis, multivariate data analysis was performed on bucketed spectra. Through an R script developed in-house, equidistant bucket tables, using a 0.02 ppm bucket width, were respectively generated for CPMG and diffusion-edited spectra in the region between 0.2 and 10 ppm (excluding the region containing the residual water signal of 4.6–4.8 ppm). Then, probabilistic quotient normalization [43] was applied to the data matrices. Random forest (RF) was used as a classification algorithm to discriminate among the groups of interest. The R package ‘Random Forest’ was used to grow a forest of 1000 trees using the default settings. To reduce the potential bias due to an unbalanced number of samples per group, the function option “sampsize” was used. The accuracy, sensitivity, specificity, and area under the receiver operating characteristic curve (AUROC) were calculated according to the standard definitions. The AUROC was assessed for significance against the null hypothesis of no prediction accuracy in the model by means of a 10^2^ randomized class-permutations test. In all spectra, the signals of a panel of 38 metabolites were unambiguously identified using the Chenomx Profiler (Version 9.0, evaluation license), the freely available database HMDB [44], and the published literature when available. Each signal was quantified via the peak integration of its spectral region. The non-parametric Wilcoxon rank–sum test was used to infer differences among the groups of interest. Robust correlations were calculated between the metabolomic and clinical data following the 10% Winsorized correlation approach using the function “wincor” of the R package “WRC2”. The *p*-values of all univariate analyses were adjusted for multiple testing using the false discovery rate (FDR) procedure with Benjamini–Hochberg correction at α = 0.05.

## Figures and Tables

**Figure 1 ijms-25-00297-f001:**
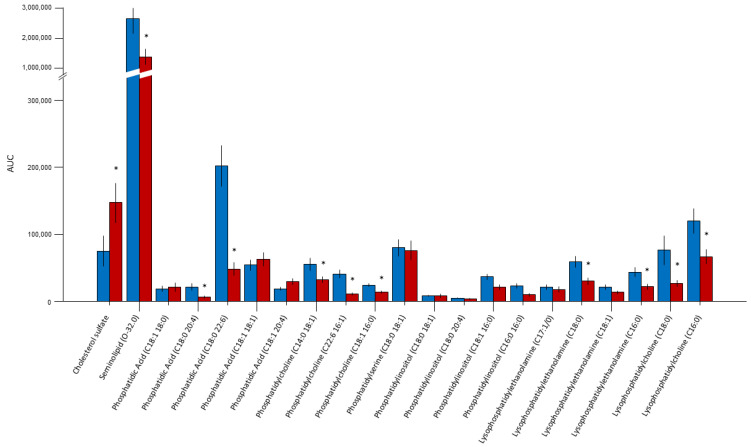
Comparative evaluation of the relative lipid composition measured by LC/MS as the AUC, standardized for the number of cells between the fertile (blue bars) and infertile (red bars) subjects. Sulfogalactosylglycerolipid (seminolipid) was the most abundant component in the membranes (72%). Cholesterol sulfate represented the second most abundant lipid (6%). PUFAs (63% of the remaining lipids in membranes), such as C20:4 (arachidonic acid) and phosphatidylcholine (C22:6 16:1), represented 12% and 25% of the total PUFAs in the sperm membranes, respectively. * *p* < 0.05.

**Figure 2 ijms-25-00297-f002:**
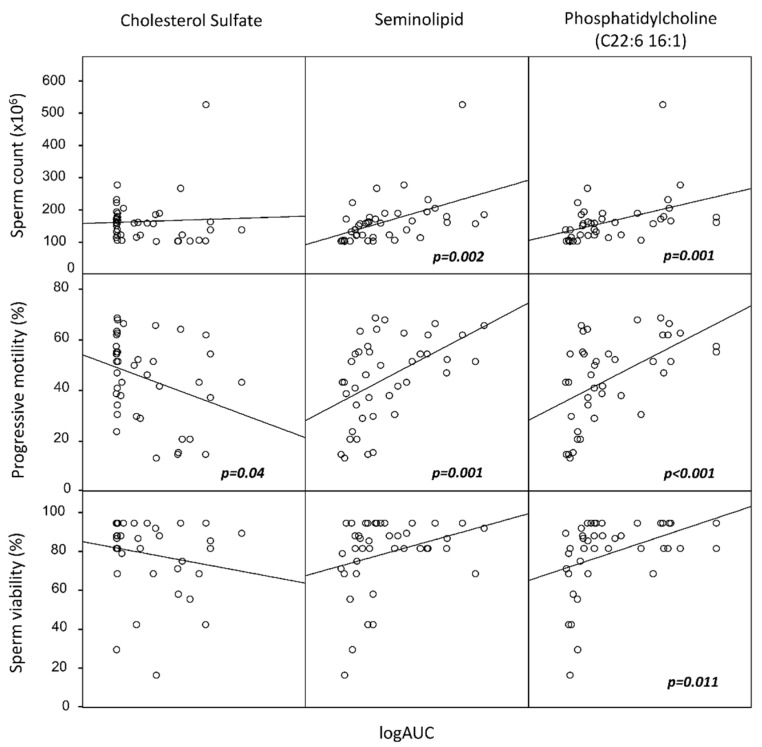
Scatter plots of correlation analysis among sperm total count, sperm motility, and sperm viability, with the lipid components identified as the most significantly associated with semen quality by principal component analysis (PCA) from the entire group of 21 lipids. In particular, seminolipid and phosphatidylcholine (C22:6 16:1) were significantly associated with improved sperm quality, whereas cholesterol sulfate was negatively associated with motility. Lipids are reported on a logarithmic scale. *p* values refer to Spearman’s non-parametric correlation analysis.

**Figure 3 ijms-25-00297-f003:**
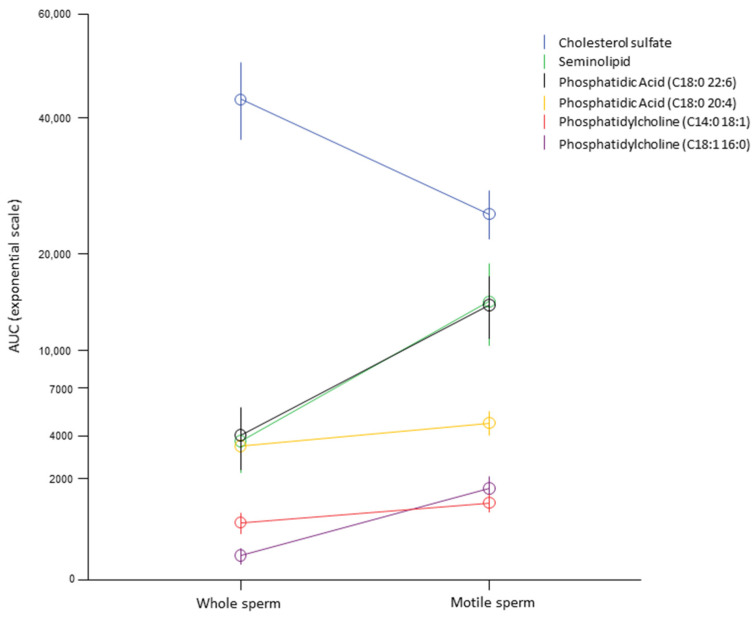
Within-subject analysis of 10 random semen from fertile subjects before and after selection for high-motility sperm by swim-up technique in order to identify the lipid association with sperm motility within subjects. Only lipids most significantly associated with sperm motility in the multivariate analysis were included. The AUC is reported as an exponential scale for clearer representation given the wide range of values of the different lipids analyzed.

**Figure 4 ijms-25-00297-f004:**
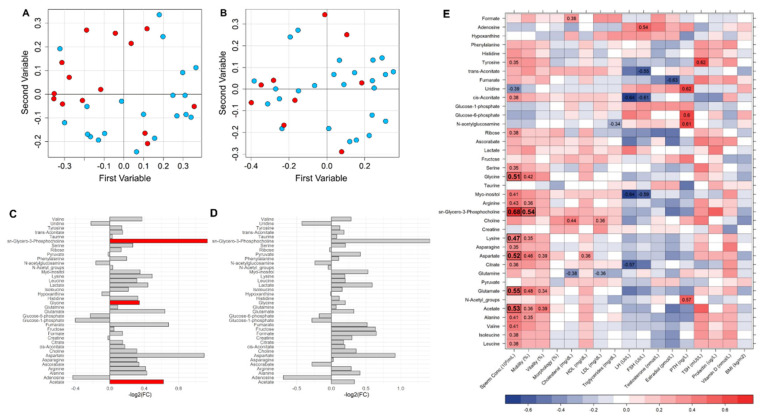
(**A**) Random forest (RF) classification plot to discriminate among the groups of interest: OLIGO patients (red dots) and CTRL (light blue dots); (**B**) ASTHENO patients (red dots) and CTRL (light blue dots). (**C**) Metabolite differences in OLIGO vs. CTRL (logarithmic scale). Grey bars represent *p*-values < 0.05 after FDR correction. Red bars identify metabolites significantly higher (*p*-value < 0.05 after FDR correction). Acetate, glycine, and sn-glycero-phosphocholine were found to be significantly higher in the seminal plasma of the oligozoospermic patients with respect to CTRL. (**D**) Metabolite differences in ASTHENO vs. CTRL (logarithmic scale). All *p*-values are <0.05 after FDR correction in the comparison of ASTHENO patients vs. CTRL. No difference between groups after correction. (**E**) Heatmap correlations between clinical and seminal plasma metabolomic parameters (red, positive correlations; blue, negative correlation). R values are reported in the plot only for statistically significant correlations (*p*-value < 0.05 after FDR correction in bold, *p*-value < 0.05 only before FDR correction not in bold). Acetate, glycine, sn-glycero-phosphocholine, lysine, aspartate, and glutamine were significantly correlated with the sperm concentration in the whole group.

**Table 1 ijms-25-00297-t001:** Clinical and seminal characteristics of fertile subjects with normal semen parameters and infertile patients with at least one alteration in semen parameters. Significant *p* values in bold.

Parameters	Fertile Group (N = 33)	Infertile Group (N = 29)	*p* Value
Age (years)	31.3 ± 8.3	32.8 ± 6.1	0.623
BMI (Kg/m^2^)	23.6 ± 2.4	24.4 ± 3.1	0.422
Luteinizing hormone (IU/L)	4.63 ± 1.65	4.52 ± 4.94	0.929
Follicle-stimulating hormone (IU/L)	4.04 ± 1.61	5.37 ± 5.41	0.320
Estradiol (pmol/L)	73.90 ± 24.10	80.20 ± 34.10	0.189
Testosterone (nmol/L)	19.79 ± 5.79	18.45 ± 7.23	0.603
Semen volume (mL)	3.14 ± 1.23	1.93 ± 1.03	**<0.001**
Total sperm count (10^6^ cells)	220.06 ± 89.42	26.88 ± 24.89	**<0.001**
Sperm concentration (10^6^ cells/mL)	81.89 ± 64.42	14.21 ± 10.69	**<0.001**
Progressive motility (%)	51.80 ± 17.98	21.22 ± 15.32	**<0.001**
Normal morphology (%)	6.44 ± 3.31	5.50 ± 3.83	0.376
Viability (%)	81.92 ± 11.94	57.08 ± 24.68	**<0.001**
Total cholesterol (mg/dL)	183.48 ± 47.93	192.11 ± 55.09	0.529
HDL (mg/dL)	55.28 ± 10.05	48.64 ± 11.45	0.612
LDL (mg/dL)	108.39 ± 41.06	105.22 ± 45.84	0.817
Triglycerides (mg/dL)	102.60 ± 82.98	131.14 ± 78.60	0.178

**Table 2 ijms-25-00297-t002:** List of all twenty-one lipids identified in the analysis and their respective ion molecular mass and retention time from LC-MS.

Lipid	Ion Molecular Mass (Da)	Main Observed Fragment	Retention Time (min)
Cholesterol sulfate	465	Fragments	8.5
Seminolipid (O-32.0)	795	539 315	9.6
Phosphatidic acid (C18:1 18:0)	701	419 283	13.0
Phosphatidic acid (C18:0 20:4)	723	419 303 283	13.4
Phosphatidic acid (C18:0 22:6)	747	419 283 327	13.5
Phosphatidic acid (C18:1 18:1)	700.7	463 281	15.1
Phosphatidic acid (C18:1 20:4)	722.8	436 303	13.4
Phosphatidylcholine (C14:0 18:1)	716.7	452 281	13.8
Phosphatidylcholine (C22:6 16:1)	850	−60 (790) 533 480 327	14.6
Phosphatidylcholine (C18:1 16:0)	804	−60 (744) 481 281	14.8
Phosphatidylserine (C18:0 18:1)	788.8	(−87) 701 419 283	15.0
Phosphatidylinositol (C18:0 18:1)	863	(−180) 683 581 420	17.1
Phosphatidylinositol (C18:0 20:4)	885	(−161) 724 600 581 419	15.4
Phosphatidylinositol (C18:1 16:0)	836	581 553 417	16.0
Phosphatidylinositol (C16:0 16:0)	810	553 417	15.1
LysoPE (P-18:0/0:0) phospho-ether lipid	464.9	403 267 196	18.6
Lysophosphatidylethanolamine (C18:0)	480	283	17.6
Lysophosphatidylethanolamine (C18:1)	478	281	17.6
Lysophosphatidylethanolamine (C16:0)	452	255	18.0
Lysophosphatidylcholine (C18:0)	508	283	20.5
Lysophosphatidylcholine (C16:0)	480	255	19.5

**Table 3 ijms-25-00297-t003:** Principal component analysis (PCA) clustering.

Matrix	Lipids Included	% of Explained Variance (Cumulative Variance)
PC 1	LysoPE(P-18:0/0:0) phospho-ether lipid Lysophosphatidylethanolamine (C18:0) Lysophosphatidylethanolamine (C18:1) Lysophosphatidylethanolamine (C16:0) Lysophosphatidylcholine (C16:0)	29.3% (29.3%)
PC 2	Seminolipid (O-32.0) Phosphatidic acid (C18:0 20:4) Phosphatidylcholine (C22:6 16:1) Phosphatidylcholine (C18:1 16:0) Phosphatidylinositol (C18:0 18:1) Phosphatidylinositol (C18:0 20:4) Phosphatidylinositol (C18:1 16:0) Phosphatidylinositol (C16:0 16:0)	16.3% (45.6%)
PC 3	Phosphatidic acid (C18:1 18:1) Phosphatidylcholine (C14:0 18:1) Phosphatidylcholine (C22:6 16:1) Phosphatidylcholine (C18:1 16:0) Phosphatidylserine (C18:0 18:1) Phosphatidylinositol (C18:1 16:0)	11.0% (56.6%)
PC 4	Phosphatidic acid (C18:0 20:4) Phosphatidic acid (C18:0 22:6) Lysophosphatidylcholine (C18:0)	8.9% (65.5%)
PC 5	Cholesterol sulfate Phosphatidic acid (C18:1 18:0) Phosphatidylserine (C18:0 18:1)	8.5% (74.0%)

## Data Availability

The data presented in this study are available upon request from the corresponding author.

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
