# Peer review of "Lipidomic Profile of Human Sperm Membrane Identifies a Clustering of Lipids Associated with Semen Quality and Function"

_ijms, 2023, doi:10.3390/ijms25010297_

Round 1

Reviewer 1 Report

Comments and Suggestions for Authors

Dear authors,

the manuscript at hand concerning lipidomic analysis of semen of fertile and infertile men is well written and clear. However, there is a major issue with the mass spectrometric identification of the lipid species!

As HILIC-separations such as the one used in the study at hand separate lipids primarily according to the head group moieties, it is highly unlikely that 18:1-18:1-PA should elute more than 1.5 min later than other PAs. Even more stunning is the identification of PAs at all, as they are typically not detectable in positive ion mode due to their acidic nature. Moreover, as they are signalling lipids, they are at least two orders of magnitude less abundant than other lipids like PCs and PEs in most cell types. I would asssume that the PA-species are formed within the ion source by in-source-decay which is often observed in ESI. Furthermore, this is to my knowledge the first time that 17:1-LPE has been identified from human samples. In fact, this lipid is often used as an internal standard as it assumed not to be naturally occuring in mammalian samples. Therefore, the retention times of the presumably identified lipids should be verified by using standards -either stable-isotope labelled or odd-chained (e.g. from Avanti)- for each lipid class identified. If the retention times of the standards do not match the ones of the identified lipids, the IDs are false and should be revisited, e.g. by MRM.

Further, there are some typos in table 2 where the retention time is missing but the m/z-ratio is given twice.

Best Regards!

Reviewer 2 Report

Comments and Suggestions for Authors

Summary of the Manuscript:

Reviewed the manuscript "Lipidomic Profile of Human Sperm Membrane Identifies a Clustering of Lipids Associated with Semen Quality and Function" by Andrea Di Nisio, et al. The study investigates the lipidomic profile of human sperm membranes and their correlation with semen quality and function. Utilizing lipidomic analysis of sperm from fertile and infertile subjects, the research identifies specific lipid clusters associated with semen parameters, including cholesterol sulfate, sulfogalactosylglycerolipid (SGG), and polyunsaturated fatty acids (PUFAs). The study highlights the independence of these sperm membrane lipids from serum lipid levels, suggesting their potential as markers of reproductive function.

Strengths

  1. Innovative Approach: The research utilizes an untargeted lipidomic analysis, providing new insights into the role of sperm membrane lipids in male fertility.
  2. Comprehensive Analysis: The study examines both fertile and infertile subjects, offering a thorough comparative analysis.
  3. Clinical Relevance: Findings highlight potential biomarkers for assessing male fertility, which could be significant for clinical diagnostics.
  4. Statistical Rigor: The use of principal component analysis and multilinear regression analysis strengthens the validity of the findings.

Weaknesses

  1. Sample Size: The study involves a limited number of subjects, which may affect the generalizability of the results.
  2. Lack of Longitudinal Data: The study is cross-sectional, limiting the ability to assess changes over time or effects of interventions.
  3. Potential Confounding Factors: While the study controls for several variables, other lifestyle or environmental factors that could influence lipid profiles were not thoroughly examined.

Recommendations

  1. Figure 1: More specific details about the types of lipids analyzed would enhance clarity. Additionally, using more distinct colors or patterns could improve visual differentiation between fertile and infertile subjects, aiding in interpretation.
  2. Figure 2: The legend is somewhat technical and could benefit from a simpler explanation of the statistical methods used, such as PCA analysis and Spearman correlation. It also lacks specific details on the lipid components identified as significant.
  3. Figure 3: The terminology used in the legend, like "swim-up sperm selection," may be too specialized for general readers. More information about why these specific samples were chosen would be useful for context.
  4. Figure 4: The detailed statistical information could be overwhelming for non-experts. Simplifying this language and providing explanations for terms like the "Random Forest (RF) model" would make it more accessible. The effectiveness of the visual representation, particularly the color coding, is uncertain without viewing the figure. Additionally, the legend could include examples or categories of the metabolites identified as significant to provide immediate insight into the key findings.

I recommend this manuscript for publication with minor revisions. The study presents valuable insights into the lipidomic profile of human sperm and its implications for male fertility, contributing significantly to the field of reproductive biology. Addressing the highlighted recommendations will further strengthen the impact of this research.

Author Response

  1. Figure 1: More specific details about the types of lipids analyzed would enhance clarity. Additionally, using more distinct colors or patterns could improve visual differentiation between fertile and infertile subjects, aiding in interpretation.

Answer: We thank the Reviewer for the helpful comment and figure colors and legend were changed accordingly, enhancing clarity of results.

  1. Figure 2: The legend is somewhat technical and could benefit from a simpler explanation of the statistical methods used, such as PCA analysis and Spearman correlation. It also lacks specific details on the lipid components identified as significant.

Answer: we thank the Reviewer and we improved figure legend.

  1. Figure 3: The terminology used in the legend, like "swim-up sperm selection," may be too specialized for general readers. More information about why these specific samples were chosen would be useful for context.

Answer: we thank the Reviewer for the helpful comment. Figure axis legend was changed to give more immediate identification of the two groups, and figure legend was improved accordingly.

  1. Figure 4: The detailed statistical information could be overwhelming for non-experts. Simplifying this language and providing explanations for terms like the "Random Forest (RF) model" would make it more accessible. The effectiveness of the visual representation, particularly the color coding, is uncertain without viewing the figure. Additionally, the legend could include examples or categories of the metabolites identified as significant to provide immediate insight into the key findings.

Answer: we thank the Reviewer and apologise for lack of clarity. Figure legend was partially re-written and we hope the new version is cleared also for non-expert readers.

Round 2

Reviewer 1 Report

Comments and Suggestions for Authors

Dear authors,

thanks for revising the manuscript accordingly, using negative ion mode sounds mcuh more reasonable with the respective identified lipids. However, I would recommend to state at least the possibility of in-source decay of more complex lipids as it cannot be excluded and could be misleading to the reader. You are right on the other hand that this is in your case not changing your conclusions.

Best Regards!

Author Response

We thank the Reviewer once again, we added this further possibility in the limits section as suggested.